# Type IV Collagen in Human Colorectal Liver Metastases—Cellular Origin and a Circulating Biomarker

**DOI:** 10.3390/cancers14143396

**Published:** 2022-07-13

**Authors:** Moa Lindgren, Gunilla Rask, Josefin Jonsson, Anette Berglund, Christina Lundin, Pär Jonsson, Ingrid Ljuslinder, Hanna Nyström

**Affiliations:** 1Department of Surgical and Perioperative Sciences/Surgery, Umeå University, SE-901 85 Umeå, Sweden; gunilla.rask@umu.se (G.R.); josefin.jonsson@umu.se (J.J.); anette.berglund@umu.se (A.B.); christina.lundin@umu.se (C.L.); hanna.nystrom@umu.se (H.N.); 2Department of Medical Biosciences/Pathology, Umeå University, SE-901 87 Umeå, Sweden; 3Department of Chemistry, Umeå University, SE-907 36 Umeå, Sweden; par.jonsson@umu.se; 4Department of Radiation Sciences/Oncology, Umeå University, SE-901 87 Umeå, Sweden; ingrid.ljuslinder@regionvasterbotten.se; 5Wallenberg Centre for Molecular Medicine, Umeå University, SE-901 87 Umeå, Sweden

**Keywords:** type IV collagen, COL IV, colorectal cancer, liver metastases, circulating biomarkers, tumor stroma, fibroblasts, CAF, MMP

## Abstract

**Simple Summary:**

Patients with colorectal liver metastases (CLM) can be cured through surgery if metastases are detected early in disease progression. Today, CLM diagnosis relies heavily on diagnostic imaging, and cheap, non-invasive, and efficiently measurable biomarkers are needed. Circulating type IV collagen (COL IV) is a potential biomarker for detecting CLM. Patients with CLM show elevated circulating levels of COL IV and increased tissue expression of COL IV in CLM tissue, which could result from enhanced production and degradation of COL IV. This study aimed to establish the cellular source behind enhanced COL IV levels, which is helpful in the evaluation of the biomarker potential of COL IV. We show that fibroblasts express COL IV both in vitro and in the stromal tissue of CLM. We also found that CLM tissue expresses COL IV-degrading proteases. Lastly, CLM patients have higher circulating COL IV levels than healthy controls.

**Abstract:**

Circulating type IV collagen (cCOL IV) is a potential biomarker for patients with colorectal liver metastases (CLM) who present with elevated levels of COL IV in both CLM tissue and circulation. This study aimed to establish the cellular origin of elevated levels of COL IV and analyze circulating COL IV in CLM patients. The cellular source was established through in situ hybridization, immunohistochemical staining, and morphological evaluation. Cellular expression in vitro was assessed by immunofluorescence. Tissue expression of COL IV-degrading matrix metalloproteinases (MMPs)-2, -7, -9, and -13 was studied with immunohistochemical staining. Plasma levels of COL IV in CLM patients and healthy controls were analyzed with ELISA. This study shows that cancer-associated fibroblasts (CAFs) express COL IV in the stroma of CLM and that COL IV is expressed in vitro by fibroblasts but not by tumor cells. MMP-2, -7, -9, and -13 are expressed in CLM tissue, mainly by hepatocytes and immune cells, and circulating COL IV is significantly elevated in CLM patients compared with healthy controls. Our study shows that stromal cells, not tumor cells, produce COL IV in CLM, and that circulating COL IV is elevated in patients with CLM.

## 1. Introduction

Colorectal cancer (CRC) is the third most common cancer worldwide [1,2], and the development of metastatic disease is the main reason for CRC-related death [3]. The liver is the most common metastatic site, and approximately 25–30% of CRC patients are diagnosed with colorectal liver metastases (CLM) [4,5,6,7]. CLM patients can achieve a long-lasting cure through surgery, offered to 15–25% of the patients [7,8,9], with a five-year overall survival of 30–50% [10,11,12]. The patients are subsequently monitored with time-consuming and expensive diagnostic imaging to detect recurrence [7]. The early diagnosis of metastases is crucial to achieving a long-lasting cure, and sensitive and cost-effective biomarkers that detect CLM at an early stage can potentially increase the number of patients eligible for curative surgery and reduce the use of diagnostic imaging.

Currently, the diagnosis of CLM relies on radiological and other imaging modalities. The tumor cell-produced circulating biomarker carcinoembryonic antigen (CEA) can be used with limited applicability [7]. CEA is most sensitive to metastatic disease, especially liver metastases, and up to 80% of CLM patients have elevated CEA [13,14,15]. However, 14–45% of patients with primary CRC also present high CEA levels [15,16,17], and the increased levels vary with tumor stage [16], tumor grade, and tumor location [18,19]. Elevations in CEA are likewise associated with other malignancies, benign conditions, and smoking [20,21,22]. The low sensitivity and specificity of CEA make it suboptimal as a biomarker [13,15,23], and its usefulness is controversial [24]. For this reason, the recommended use of CEA is limited to monitoring patients after surgery to detect local and distant recurrence [7,25]. Biomarker research has, over the years, changed strategy from the unilateral focus on the tumor cell compartment to a more synoptical view by also studying the stromal compartment.

The tumor stroma constitutes the non-malignant part of cancer and comprises vasculature, extracellular matrix (ECM), and non-malignant cells, including cancer-activated fibroblasts (CAFs). Cancer progression entails the disruption of tissue homeostasis, accompanied by increased degradation of the ECM by matrix metalloproteinases (MMPs) and increased ECM and collagen deposition, called desmoplasia [26,27], and this remodeling of the microenvironment aids cancer progression [26,27,28]. MMPs are essential for metastatic disease, and several MMPs are associated with CRC and CLM [29,30,31,32,33,34,35,36]. Increased collagen deposition facilitates cell proliferation and migration [26], and in CLM tissue, collagen deposition correlates with treatment resistance and a poor prognosis [37,38]. In addition, elevated levels of circulating collagen are observed in both the urine [39] and plasma of CLM patients [40,41,42,43]. CAFs are believed to be the primary source of desmoplasia, although tumor cells may produce ECM [26,44,45,46,47,48,49].

Type IV collagen (COL IV) is a stromal protein identified as a potential biomarker in patients with CLM [40,41,50,51]. COL IV is generally expressed in all basement membranes of the body [52,53], and increased expression is observed in the stroma of CLMs [40,54]. Circulating levels of COL IV (cCOL IV) are also increased in patients with CLM and correlate with a poor prognosis and hepatic tumor burden [40,41,42]. In addition, MMP-mediated fragments of COL IV are elevated in serum of metastatic CRC patients compared with patients without metastases [55]. An animal study showed that COL IV expression in murine lung carcinoma cells correlates with liver-metastatic potential, and liver-metastatic tumor cells avoid anoikis by binding to endogenous COL IV via integrins [54]. Cell experiments on breast cancer cells and pancreatic cancer cells show that contact between tumor cells and COL IV induces the expression of proteases [56,57] and that tumor cells utilize endogenous COL IV for cell growth, migration, and the avoidance of apoptosis [58]. Furthermore, elevated levels of COL IV in both circulation and tissue are also observed with other cancer types [49,59,60]. We have previously shown that high levels of cCOL IV in metastatic breast cancer correlate to a poor prognosis [59]. We have also demonstrated that the sensitivity in detecting CLM can be improved by combining cCOL IV with the conventional biomarker, CEA [41].

As previously described, cancer-related disruption of tissue homeostasis aids cancer progression, including increased COL IV levels that seem essential for cell growth, migration, and the liver-metastatic potential. Thus, elevated cCOL IV levels could result from both stromal degradation and deposition. Nevertheless, the cellular origin of cCOL IV in CLM patients is not known, nor is the mechanism explaining why they present with increased circulating levels.

Circulating COL IV might be a marker of metastatic disease. However, it is crucial to establish the origin of cCOL IV elevations in CLM, as metastases such as CLM can present with heterogeneous stromal architecture [61]. It is not known whether COL IV is produced by the tumor cells, the stromal cells, or a combination of both compartments. Increased knowledge about tumor stroma remodeling, including deposition and degradation, can help us understand when and how we best utilize cCOL IV as a biomarker. A sensitive circulating biomarker that can detect CLM at an early stage could, at least partly, replace time-consuming and expensive imaging modalities and increase the number of patients eligible for surgery—the only lasting cure for CLM. In addition, cCOL IV might be valuable in monitoring metastatic disease during chemotherapy.

The overarching goal of this study was to increase our knowledge concerning COL IV in CLM patients. The primary aim was to determine the cellular source of COL IV in CLM tissue and verify the results in relevant cell lines. Secondly, we aimed to characterize tissue expression of known COL IV-degrading MMPs (MMP-2, -7, -9, and -13) in CLM. The third aim was to evaluate the potential of cCOL IV as a diagnostic biomarker for resectable CLM patients. Lastly, we aimed to investigate the prognostic value of cCOL IV for resectable CLM patients.

## 2. Materials and Methods

### 2.1. Ethics Statement

Informed consent was obtained from all the patients, and the Swedish Ethical Review Board approved this study (DNR 09-175M) in compliance with the Helsinki Declaration of 1975.

### 2.2. Patients and Sample Collection for Tissue Analysis

Archival formalin-fixed paraffin-embedded (FFPE) tissue from 30 patients who underwent liver surgery at Umeå University Hospital, Umeå, Sweden, between 1998 and 2009, was included in this study. One representative tissue block of viable CLM with adjacent liver parenchyma was selected for each case. Five-micrometer sections were cut with a microtome (Leica RM 2165, Leica Microsystems, Wetzlar, Germany). Tissue from these patients was included in a previously published study [50]. Clinical data were collected from patient charts.

RNA and protein expression in the tissue sections was evaluated by three researchers: a pathologist (G.R.), a Ph.D. student (M.L.), and a researcher/surgeon (H.N.), all well-acquainted with methods of histopathology.

### 2.3. Patients and Sample Collection for Plasma Analysis

Circulating COL IV was measured in preoperative plasma samples (*n* = 138) from patients who underwent liver surgery for CLM at Umeå University Hospital between 2009 and 2015. All the patients were included in the biobank prospectively, but they were analyzed retrospectively in accordance with the REMARK guidelines. The inclusion criteria for this study were as follows: patients admitted for liver surgery with a curative intent and available plasma samples stored in the upper gastrointestinal biobank in Umeå, with plasma samples being collected from the patients within 50 days before surgery. The exclusion criteria for this study were: no available plasma samples as stated above, samples collected more than 50 days prior to surgery, the patient did not undergo planned liver resection, or the patient passed away shortly (<1 month) after resection due to non-cancer-related disease. Blood samples were drawn by peripheral venous puncture and plasma was aliquoted and stored at −80 °C until analysis. Clinical data were collected from patient charts. Plasma samples from 118 healthy individuals were collected within the Västerbotten intervention program [61] and used as controls.

### 2.4. Expression of Type IV Collagen in Colorectal Liver Metastases and Cell Lines

#### 2.4.1. Protein Expression of COL IV in Colorectal Liver Metastases

Staining for the protein expression of COL IV had previously been performed on 30 sections, as follows: Anti-human type IV collagen (protease, rabbit polyclonal, MP Cappel 1:75, incubated for 32 min at 37 °C; Fischer Scientific, Gothenburg, Sweden). A secondary antibody linked to peroxidase followed by diaminobenzidine tetrahydrochloride (DAB) as a chromogen was used. Negative control sections were incubated with secondary antibodies only. This staining was performed on the Ventana Benchmark automated immunostainer (Ventana Medical Systems, Tucson, AZ, USA) [50].

#### 2.4.2. RNA Expression of COL IV in Colorectal Liver Metastases

To assess the cellular origin of COL IV in CLM, COL IV RNA expression was analyzed with in situ hybridization (ISH). Paraffinized CLM tissue (*n* = 15) with the strong protein expression of COL IV was selected for analysis and the RNA expression of COL IV was analyzed with a ViewRNA^TM^ ISH Tissue 1-plex assay (QVT0050, ThermoFisher Scientific, Waltham, MA, USA), with a ViewRNA^TM^ chromogenic signal amplification kit (QVT0200, ThermoFisher Scientific, Waltham, MA, USA) and ViewRNA^TM^ Tissue probe COL4A1 (VX-01, ThermoFisher Scientific, Waltham, MA, USA), according to the manufacturer’s instructions. Sections were deparaffinized in xylene, followed by incubation for 20 min in pretreatment solution preheated to 90 °C. The sections were treated with protease (1:100) and incubated in a hybridization oven at 40 °C for 20 min before fixation and subsequent probe hybridization. The probe was diluted (1:50) in prewarmed diluent (40 °C) and added to tissue sections, followed by incubation in a hybridization oven for 3 h at 40 °C. The sections were left in storage buffer overnight at room temperature. Sections were treated with preamplification solution (1:100) and incubated in the hybridization oven for 25 min at 40 °C, followed by treatment with amplification solution (1:100) and incubation in the oven for 15 min at 40 °C. The sections were subsequently incubated with working labeled probe solution (1:1000) and fast red substrate, according to instructions, followed by counterstaining with Gill’s hematoxylin for 30 s. Sections were mounted with ADVANTAGE^TM^ mounting medium (Innovex Biosciences, Richmond, CA, USA).

The RNA expression was analyzed as cytoplasmic positivity in tumor cells, hepatocytes, and/or stromal cells. Positive cells were later identified by morphological evaluation and immunohistochemical analysis.

#### 2.4.3. Expression of CAF Markers in Colorectal Liver Metastases

To verify the potential results of the ISH and to establish whether CAFs are present in the COL IV-rich stroma of CLM, tissue sections (*n* = 30) were stained for CAF-specific markers. Tissue from the same metastases as analyzed with ISH (*n* = 15) and an additional 15 CLMs were analyzed. Primary antibodies and dilutions were used as follows: sheep-anti-FAP (1:100, AF3715, R&D Systems, Minneapolis, MN, USA) and rabbit-anti-α-SMA (1:200, ab5694, Abcam, Cambridge, UK). Both antibodies were polyclonal. A secondary antibody linked to peroxidase followed by diaminobenzidine tetrahydrochloride (DAB) as a chromogen was used. Negative control sections were incubated with secondary antibodies only. This staining was performed on the Ventana Benchmark automated immunostainer (Ventana Medical Systems, Tucson, AZ, USA) for anti-α-SMA and manually for anti-FAP. With manual staining, the sections were treated with citrate buffer in a pressure cooker for 1 h.

#### 2.4.4. Expression of COL IV in Cell Lines

##### Cell Lines and Cell Culturing

Three fibroblasts cell lines were used (all from American Type Culture Collection (ATCC), Manassas, VA, USA): CCD-18Co (CRL-1459) and CCD-112CoN (CRL-1541) are fibroblast cell lines derived from normal human colon, and WS1 (CRL-1502) is a fibroblast cell line derived from normal human skin.

Three cancer cell lines were used (all from ATCC, Manassas, VA, USA): HT-29 (HTB-28) and SW480 (CCL-228) are cell lines derived from human primary colorectal adenocarcinoma, and LoVo (CCL-229) is a cell line of human metastatic colorectal adenocarcinoma derived from the supraclavicular region.

CCD-18Co and HT-29 were cultured in Dulbecco’s Modified Eagle Medium (DMEM, Gibco, Waltham, MA, USA), supplemented with 10% fetal bovine serum (FBS, Gibco, Waltham, MA, USA). CCD-112CoN and WS1 cells were cultured in Eagle’s Minimum Essential Medium (EMEM, ATCC, Manassas, VA, USA), supplemented with 10% fetal bovine serum (FBS, Gibco, Waltham, MA, USA). SW480 cells were cultured in Leibovitz’s L-15 Medium (Gibco, Waltham, MA, USA), supplemented with 10% fetal bovine serum (FBS, Gibco, Waltham, MA, USA). LoVo cells were cultured in Dulbecco’s Modified Eagle Medium Nutrient Mixture F-12 (DMEM/F-12, Gibco, Waltham, MA, USA), supplemented with 10% fetal bovine serum (FBS, Gibco, Waltham, MA, USA). All the cell lines except SW480 were grown in an incubator at 37 °C in an atmosphere with 5% CO_2_. SW480 were grown in an incubator at 37 °C but without 5% CO_2_.

##### Analysis of COL IV Expression in Cell Lines

The expression of COL IV in fibroblasts and tumor cells was assessed by immunofluorescence (IF) analysis of cells cultured in recommended cell media on Falcon™ Culture Slides (BD Biosciences, Erembodegem, Belgium). The primary antibody used was rabbit anti-human type IV collagen (1:100, AB748, Merck KGaA, Darmstadt, Germany) and, as a secondary antibody, donkey anti-rabbit FITC (1:100, ImmunoResearch Laboratories, Inc., West Grove, PA, USA) was used. Sections were mounted with medium containing DAPI (Vectashield, Vector Laboratories, Inc., Burlingame, CA, USA). As a negative control, cells were incubated with secondary antibodies only.

All cell experiments were run in duplicate with a negative control, where the primary antibody was excluded. Each experiment was repeated three times. Cell lines were regarded as COL IV-expressing cell lines if the cells showed positive IF staining.

### 2.5. Expression of Type IV Collagen-Degrading Proteases in Colorectal Liver Metastases

The expression of known COL IV-degrading MMPs was assessed through the immunohistochemical staining (IHC) of CLM tissue (*n* = 30) and primary antibodies, and the dilutions were as follows: mouse-anti-MMP2 (1:50, MAB13431, Merck KGaA, Darmstadt, Germany), mouse-anti-MMP7 (1:100, MAB9071, R&D Systems, Minneapolis, MN, USA), mouse-anti-MMP9 (1:200, MAB911, R&D Systems, Minneapolis, MN, USA), and mouse-anti-MMP13 (1:20, MAB511, R&D Systems, Minneapolis, MN, USA). All the antibodies were monoclonal. A secondary antibody linked to peroxidase, followed by diaminobenzidine tetrahydrochloride (DAB) as a chromogen, was used. Negative control sections were incubated with secondary antibodies only. This staining was performed on the Ventana Benchmark automated immunostainer (Ventana Medical Systems, Tucson, AZ, USA) for anti-MMP-7 and MMP-9 and manually for anti-MMP-2 and MMP-13. With manual staining, the sections were treated with citrate buffer in a pressure cooker for 1 h. One section stained with mouse-anti-MMP-13 was derived from a different area, but the same metastasis, compared with sections analyzed for MMP-2, -7, and -9.

MMP-2 scoring was based on the intensity of cytoplasmic granular staining in hepatocytes and tumor cells, as follows: 0, no positivity in any cell; 1, mild staining in some to many cells or strong staining in a few cells; 2, moderate to strong staining in many cells. The scoring of stromal cells was based on the amount of stained immune cells, as follows: 0, no positivity in any cell; 1, a few to some stained immune cells; 2, many stained immune cells.

MMP-7, -9, and -13 scoring were based on the intensity of cytoplasmic granular staining in hepatocytes and tumor cells, as follows: 0, no positivity in any cell; 1, staining in some cells; 2, staining in many cells. The scoring of stromal cells was based on the amount of stained immune cells, as follows: 0, no positivity in any cell; 1, some stained immune cells; 2, many stained immune cells. In the event of different results, the sections were re-evaluated, and a consensus was reached.

### 2.6. Circulating Type IV Collagen in Patients with Colorectal Liver Metastases

The circulating levels of COL IV in CLM patients and controls were measured in plasma with a Collagen IV EIA kit (Sensitivity 0.1 ng/mL, Argutus Medical, Dublin, Ireland), a sandwich-ELISA kit with two monoclonal antibodies directed against the 7S and the collagenous domain of COL IV. Samples were run in duplicate and analyzed according to the manufacturer’s procedure. The difference between duplicate samples was not allowed to be more than 12.5%.

### 2.7. Statistical Analysis

Circulating COL IV levels in CLM patients were compared with cCOL IV levels in healthy controls using the Mann–Whitney test. Patients that were the subject of repeat resections were included as a new event. Differences in gender and age between patients and controls were analyzed with the chi-square and Mann–Whitney tests, respectively.

The effects of age and gender on the level of cCOL IV were analyzed separately for patients and controls using multiple linear regression. Multiple linear regression was also used to assess whether gender and age affected the difference in cCOL IV between patients and controls.

The effects of the size of the largest metastasis (mm), disease-free interval (months), age (years), and number of liver metastases on cCOL IV levels were analyzed with Pearson’s correlation. The effect of preoperative chemotherapy (yes vs. no), TNM stage (1–3 vs. 4), size of the largest metastases (<5 cm vs. >5 cm), number of liver metastases (<1 vs. >1), node-positive primary tumor, localization of primary tumor (colon vs. rectum), primary tumor present at time of blood sampling, gender, extrahepatic metastatic disease, age (<70 y/o vs. >70 y/o), and disease-free interval (<12 months vs. >12 months) on cCOL IV levels were analyzed with a t-test. Any potential relationship between prognostic factors was analyzed with a t-test and the chi-square test. Missing values were omitted for analysis.

Receiver operating characteristic (ROC) analysis was applied to calculate the area under the curve (AUC) to establish the potential of cCOL IV to discriminate between patients and controls. Logistic regression was used to adjust for age and ROC analysis was repeated. To find the optimal cutoff point, we used the approach known as “the point closest to (0, 1) corner in the ROC plane”, which defines the optimal cutoff point as the point minimizing the Euclidean distance between the ROC curve and the (0.1) point.

Kaplan–Meier survival curves were used to analyze overall survival (OS) for CLM patients. For survival analysis, the plasma cohort was divided into two groups based on the median cCOL IV value. For patients who underwent repeat surgery due to a recurrence, only the first event was included in the prognostic analysis (*n* = 128). Survival data were collected from the Swedish death registry. The study endpoint was death from any cause or the end of the current follow-up (19 October 2021). The log-rank test was used to determine differences between the curves.

GraphPad Prism version 9 (GraphPad Software, San Diego, CA, USA) and MATLAB version 9.11.0.1769968 (R2021b) (The Mathworks, Inc., Natick, MA, USA) were used. A *p*-value of <0.05 was considered statistically significant.

## 3. Results

### 3.1. Patient Characteristics of Tissue Cohort

To assess the tissue expression of COL IV, α-SMA, FAP, and MMPs, tissue sections from CLMs were analyzed. Patient characteristics are presented in Table 1.

### 3.2. Type IV Collagen Is Expressed by CAFs in the Stroma of Colorectal Liver Metastases (CLM)

IHC analysis of CLM (*n* = 30) showed that COL IV was strongly expressed in the stroma of all metastases (Figure 1A,B), in the space of Disse, the liver capsule, the portal tracts, and the basement membranes of blood vessels. No COL IV expression was observed in tumor cells (TC) or hepatocytes. To establish the cellular origin of COL IV in CLM, tissue sections of CLM (*n* = 15) were analyzed with ISH. COL IV RNA was observed in fibroblasts of 14 CLMs (Figure 2A–C). The ISH-positive cells were morphologically verified as fibroblasts and were also positive for CAF markers α-SMA and FAP (Figure 2D,E). ISH positivity was also observed in a smaller number of endothelial cells. We did not observe any difference in COL IV RNA expression when comparing stromal expression in the tumor–stroma interface with the central parts of the tumor.

### 3.3. Type IV Collagen Is Produced by Fibroblasts In Vitro but Not by Colorectal Cancer Cells

The expression of COL IV in vitro was assessed by IF analysis of three CRC cell lines and three fibroblast cell lines. The results showed that COL IV is only produced in vitro by fibroblasts derived from the colon. COL IV was clearly expressed by CCD-18Co and CCD112-CoN (see Appendix A). CCD-18Co was positive for COL IV for all experiments, whereas CCD112-CoN was positive in two of three experiments. WS1 was negative for COL IV in all experiments and all CRC cell lines were negative (see Appendix A).

### 3.4. MMP-2, -7, -9, and -13 Are Expressed in CLM

The expression of COL IV-degrading proteases in CLM was analyzed with IHC on FFPE sections from CLM patients (*n* = 30). The expression of MMP-2, -7, -9, and -13 was observed in hepatocytes and immune cells of all sections (*n* = 30), as illustrated in Figure 3. The score of MMP expression is summarized in Table 2 and Figure 4.

MMP-2: MMP-2 staining in immune cells was most often scored as 1 (*n* = 20, 66.7%), referring to few or some stained immune cells. The staining of hepatocytes was mainly scored as 2 (*n* = 24, 80%), referring to moderate to strong staining in many cells. Positive staining in tumor cells was observed in 5 (16.7%) sections (score 1), referring to mild staining in some to many cells or strong staining in a few cells.

MMP-7: MMP-7 staining was mostly scored as 1 in both immune cells (*n* = 25, 83.3%) and hepatocytes (*n* = 24, 80%), referring to some stained cells. Expression in tumor cells was also scored as 1 in 10 sections (33.3%), but the majority did not express MMP-7 (*n* = 19, 63.3%).

MMP-9: All sections were positive for MMP-9 expression in immune cells and the majority were scored as 1 (*n* = 17, 56.7%). Positive expression in hepatocytes was scored as 1 (*n* = 17, 56.7%). No tumor cells expressed MMP-9.

MMP-13: MMP-13 staining was primarily scored as 1 in both immune cells (*n* = 17, 56.7%) and hepatocytes (*n* = 22, 73.3%). No tumor cells expressed MMP-13.

### 3.5. Patient Characteristics of Plasma Cohort

The levels of cCOL IV were measured in preoperative plasma samples (*n* = 138) from 128 patients with CLM undergoing liver surgery. The flowchart in Figure 5 illustrates patients included in and excluded from the cohort for the analysis of cCOL IV. One hundred and four patients were excluded due to no available blood samples. Two patients were excluded after plasma analysis: one was excluded due to death shortly after surgery (myocardial infarction), and one did not undergo planned liver surgery due to irresectable disease. The final inclusion of the plasma CLM cohort was *n* = 138. Nine patients in the cohort underwent repeat liver resections (eight patients underwent two liver resections and one patient underwent three). As a result, the analyzed cohort was based on 128 unique patients and 138 samples. The majority of patients were diagnosed with CLM less than 12 months after the primary diagnosis (*n* = 112, 81%). Of the 138 patients, 39 had their primary tumor present at the time of liver surgery (28%). The median follow-up time for the cohort was 52 months, and 97 patients (70.3%) died during follow-up. The 5-year overall survival after liver surgery was 48.6%. Patient characteristics are presented in Table 3.

### 3.6. Circulating Level of COL IV Is Elevated in Patients with CLM

We analyzed the biomarker potential of COL IV by measuring the level of cCOL IV in CLM patients (*n* = 138) and healthy controls (*n* = 118). CLM patients had significantly higher levels of cCOL IV (174.1 ng/mL (69.9–483.1)) when compared with healthy controls (101.2 ng/mL (42.4–239.2)) (*p* < 0.0001) (Table 3, Figure 6). There was a significant difference between CLM patients and controls in terms of gender (*p* = 0.0049) and age (*p* < 0.0001).

Separate regression models for controls and patients showed that the level of cCOL IV in controls increased with age (*p* < 0.0001), but this was not the case for CLM patients (*p* = 0.1295). Gender did not affect the level of cCOL IV for either controls (*p* = 0.8109) or patients (*p* = 0.6072). Multiple linear regression analysis of age and case-control status in patients and controls combined showed that age (*p* = 0.0011) and case-control (*p* < 0.0001) significantly affected cCOL IV levels. Age thus had an effect on the model, but the effect of case-control status was greater.

A significant correlation was found between the levels of cCOL IV in CLM patients and the size of the largest metastasis (*p* = 0.0338), preoperative chemotherapy (*p* = 0.0023), and TNM stage (*p* = 0.0378) (see Appendix A for results). Additionally, preoperative chemotherapy was related to the size of the largest metastasis (*p* = 0.0032), as well as TNM stage (*p* = 0.0012). There was no correlation between the levels of cCOL IV and any of the other prognostic factors of CLM (see Appendix A).

### 3.7. Circulating Level of COL IV Distinguishes CLM Patients from Controls

The ROC analysis revealed that cCOL IV is able to distinguish CLM patients from healthy controls (AUC 0.8791, S.E = 0.02, *p* < 0.0001) (Figure 7). The AUC was improved when adjusting for age with logistic regression (AUC 0.9027, S.E = 0.018, *p* < 0.0001). The optimal cutoff, sensitivity, and specificity are presented in Table 4. The optimal cutoff for this cohort was 135.2 ng/mL, with a sensitivity of 74.6% and a specificity of 83.9%. The sensitivity increased to 83.3% and the specificity decreased to 80.5% when adjusting for age.

### 3.8. Levels of Circulating COL IV Do Not Correlate with Survival for CLM Patients

No correlation between high cCOL IV levels and shorter survival time was observed (*p* = 0.9983) (data not shown). The median survival time in the group with low cCOL IV (<174.1 ng/mL) was 58 months, compared with 46 months in the group with high cCOL IV levels (>174.1 ng/mL). The median time from primary tumor to the development of metastatic disease in the CLM cohort was 0 months (range 0–57 months) and the median follow-up time was 52 months.

## 4. Discussion

No optimal circulating biomarkers for CLM exist, and CLM diagnosis and disease monitoring rely mainly on diagnostic imaging. Therefore, a non-invasive, low-cost, and highly sensitive novel biomarker that can detect CLM early in disease progression would have high clinical value. A likely source of new biomarkers for metastatic disease is the tumor stroma, the product of tumor-promoting microenvironment remodeling. Further, understanding the origin and turnover of such a biomarker will aid future studies on how to best investigate and use this biomarker. One potential biomarker for CLM patients is COL IV.

Our RNA expression analysis shows that CAFs constitute the cellular origin of COL IV in CLM. The co-localization of FAP and α-SMA expression and the morphological evaluation confirmed these findings. Tumor cells of CLM tissue and CRC cell lines expressed no COL IV. One limitation of this analysis is that RNA expression does not necessarily coincide with secreted protein since COL IV is modified by posttranslational modifications [53]. However, the analyses of COL IV, FAP, and α-SMA were performed on tissue sections from the same metastases as the RNA expression analysis, confirming COL IV expression by CAFs in the stroma of CLM. Our result agrees with observations of COL IV expression in healthy and injured livers, where hepatic stellate cells, portal fibroblasts, and myofibroblasts produce most of the ECM proteins [62].

Based on the reported ability of tumor cells to produce ECM, we hypothesized that not only CAFs but also tumor cells might contribute to COL IV production in CLM [26,49,63,64,65]. Moreover, a previous study by Nystrom et al. showed intense expression of COL IV close to tumor cells in the primary tumor of patients with CRC that subsequently developed CLM [50]. Our study did not show COL IV expression by tumor cells in CLM, which was in agreement with our expression analysis of cell lines but contrasts with a study by Ikeda et al. [65]. Ikeda et al. analyzed RNA extracted from CRC cells (HT-29, SW480, and LoVo) and showed COL IV expression by all cell lines [65]. Different methods could explain this discrepancy. The culturing of CRC cell lines on Matrigel by Ikeda et al. likely reflects the impact of the surrounding microenvironment on tumor cell behavior. We showed that colon-derived fibroblasts, but not tumor cells, produce COL IV when cultured in cell media alone. The tumor microenvironment may stimulate COL IV production in tumor cells, and expression might change with cancer progression, where tumor cells in metastases might switch from endogenous production to expression through stimulation of surrounding cells. Tumor cells appear to cause fibroblast infiltration in a COL IV-PDGF-A-dependent manner, suggesting that tumor cells produce COL IV themselves early in metastatic progression. Subsequently, fibroblasts are activated, accumulate at the site, and replace tumor cells as primary collagen producers in the metastases [64].

The progression of primary CRC is associated with increased expression of MMPs [31,32,33,34,35,36], and ECM degradation likely contributes to the levels of fragmented COL IV observed in CLM [55]. Studies describing MMP expression in patients with CLM are lacking, so we analyzed the expression of known COL IV-degrading MMPs in CLM tissue. Our results show that MMP-2, -7, -9, and -13 are mainly expressed by immune cells and hepatocytes, and no tumor cells were positive for MMP-9 or -13. These MMP results contrast with another study where tumor cells of CLM expressed MMP-2 and -9 [31]. In our analysis, MMP-9 was predominately expressed by immune cells in CLM, which agrees with a publication by Illeman et al. [35] but is also expected since MMP-9 is regularly expressed by neutrophils [66]. Several MMPs, including MMP-2 and -9, are mainly produced by stromal cells, and tumor cells are believed to stimulate MMP expression through paracrine signaling [36,67,68,69]. Our analysis is limited by the absence of immunomarkers, and immune cells are identified solely by cell morphology. Further, the activity status of the MMPs and their effect on COL IV degradation is unclear. However, this is a step towards understanding the complex microenvironment of CLM. Future analyses should focus on active MMPs and their potential correlation to circulating levels of fragmented collagen.

The result of our ELISA analysis validates previous studies [40,41] by showing that CLM patients present with elevated levels of cCOL IV. We also observed an increase in cCOL IV with age in healthy controls, which was not observed in CLM patients. Compared to age, the case-control status correlated more strongly with cCOL IV, and the correlation strengthened when we corrected for age. Our results indicate that, regardless of age, there is a difference in cCOL IV between healthy controls and CLM patients. Additionally, the level of cCOL IV appeared to be increased in patients with high TNM stage and large metastases and in those who received preoperative chemotherapy. These parameters correlate to each other, probably due to patients with a high TNM stage and large metastases being more likely to receive chemotherapy than patients with a lower TNM stage and smaller tumors. The levels of cCOL IV likely reflect the tumor burden, which is coherent with previous observations [40]. The elevated levels of cCOL IV indicate that cCOL IV has potential diagnostic value as a biomarker.

The calculated optimal cutoff for cCOL IV for this cohort was 135.2 ng/mL, with a sensitivity of 74.6% and a specificity of 83.9%, which is higher than previously reported for CLM patients undergoing liver surgery [41]. Of course, different cohorts give rise to different cutoffs, which explains why a potential biomarker must be analyzed in large cohorts before a representative cutoff can be determined.

We did not observe any prognostic value of cCOL IV, as has been observed for CRC [51], pancreatic cancer [49], metastatic breast cancer [59], and CLM (combination of cCOL IV and CEA) [41]. The discrepancy could result from methodological differences or the state of the disease since some of these studies have investigated cCOL IV in palliative patient settings. In this study, we only investigated patients who were the subject of surgery with curative intent. We hypothesize that the prognostic value of cCOL IV in this study likely reflects the tumor burden, explaining why the prognostic information of cCOL IV is lost when surgery removes all tumor tissue.

We showed in this study that stromal cells, not tumor cells, produce COL IV and COL IV-degrading proteases in CLM. A combination of circulating biomarkers derived from both the stromal and the tumor compartments is likely the best option for enabling early detection of CLM. We have previously reported that, when combining biomarkers from the tumor and stromal departments, the sensitivity in detecting CLM is improved, as it is when combining cCOL IV with CEA [41]. Therefore, future studies should analyze the diagnostic biomarker potential of combinations of stromal cell- and tumor cell-produced proteins in CLM patients.

This study, to our knowledge, is the first to demonstrate that CAFs produce COL IV in CLM. Establishing the cellular origin is also a step forward in understanding the potential tumor-promoting mechanism of COL IV. Future studies should aim to further investigate the mechanisms of COL IV and its role in the metastatic process.

The focus of this paper was, however, to further investigate COL IV as a potential biomarker in patients with CLM and to analyze the origin of the biomarker and its known degrading enzymes in CLM tissue.

## 5. Conclusions

This study showed that CAFs express COL IV in human CLM tissue and that the primary source of COL IV-degrading MMP expression in CLMs seems to be stromal cells and hepatocytes. The tumor cells of CLMs did not express COL IV, and only limited MMP production was observed. Our in vitro analysis showed COL IV expression by colon fibroblasts but not by CRC cell lines. Lastly, we found that CLM patients have higher levels of cCOL IV than healthy controls and that cCOL IV measurement can significantly distinguish CLM patients from healthy controls.

## Figures and Tables

**Figure 1 cancers-14-03396-f001:**
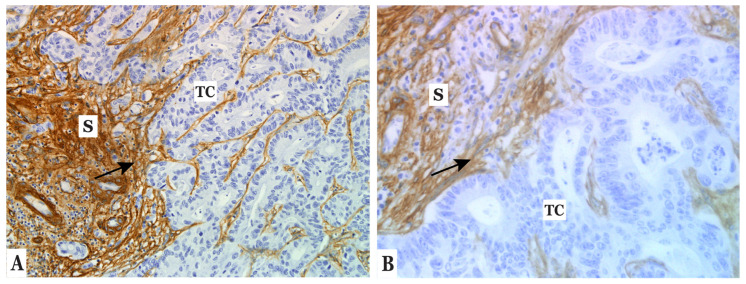
Immunohistochemical staining showing COL IV expression (brown) in stroma (arrow, S) of CLM. (**A**) Magnification ×20 and (**B**) ×40. Abbreviations: TC = tumor cells, S = stroma.

**Figure 2 cancers-14-03396-f002:**
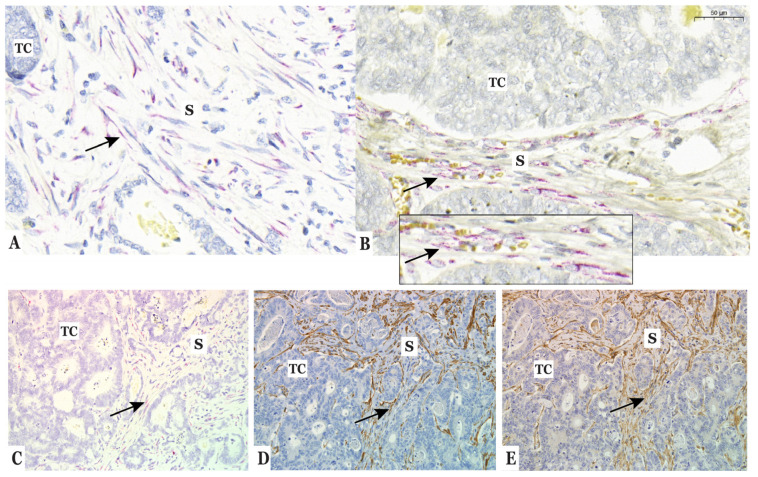
Type IV collagen (COL IV) is expressed by CAFs in colorectal liver metastases (CLM). (**A**–**C**) The RNA expression of COL IV was analyzed with in situ hybridization (ISH) and COL IV (pink) was expressed by fibroblasts (arrow) in the stroma (S) of CLM. (**C**–**E**) To validate COL IV expression by fibroblasts, consecutive tissue sections were immunohistochemically stained for CAF markers (**D**) α-SMA and (**E**) FAP. The expression (brown) of α-SMA and FAP in the stroma (arrow) of CLM was compared with the (**C**) RNA expression of COL IV (pink). (**A**,**B**) Magnification ×40 and (**C**–**E**) ×20. Extra unspecified magnification to visualize COL IV RNA-positivity. Abbreviations: TC = tumor cells, S = stroma.

**Figure 3 cancers-14-03396-f003:**
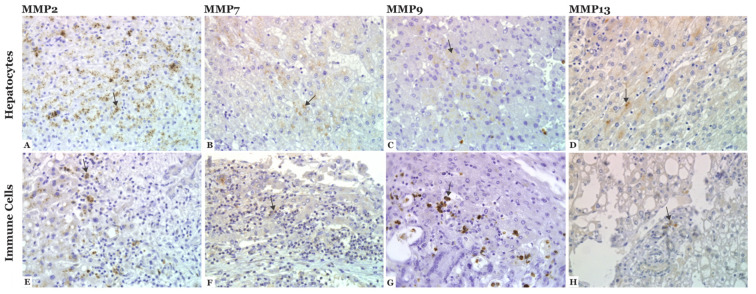
Immunohistochemical staining of matrix metalloproteinases (MMP) (brown) in colorectal liver metastases (CLM) (*n* = 30). (**A**–**D**) The analysis shows that MMP-2, -7, -9, and -13 are mainly expressed in hepatocytes (arrow) and (**E**–**H**) immune cells (arrow). Magnification ×40.

**Figure 4 cancers-14-03396-f004:**
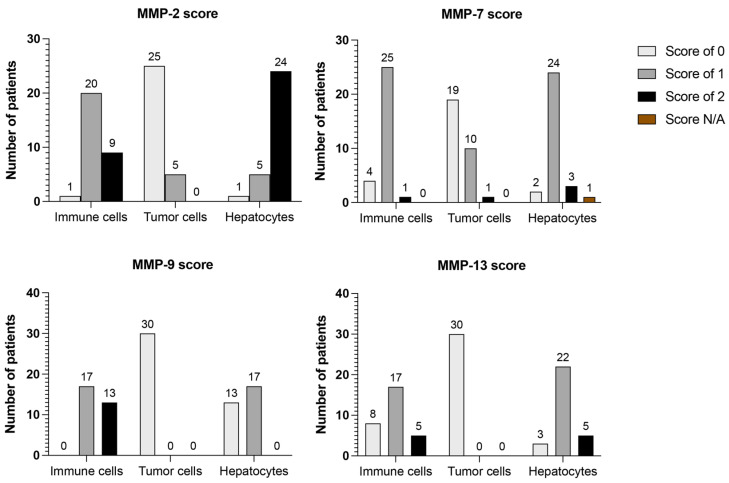
Box plots of matrix metalloproteinases (MMP) scoring in colorectal liver metastases (CLM) (*n* = 30). The analysis shows that MMP-2, -7, -9, and -13 are mainly expressed in hepatocytes and immune cells. MMP-2 scoring was based on the intensity of cytoplasmic granular staining in hepatocytes and tumor cells, as follows: 0, no positivity in any cell; 1, mild staining in some to many cells or strong staining in a few cells; 2, moderate to strong staining in many cells. The scoring of stromal cells was based on the amount of stained immune cells, as follows: 0, no positivity in any cell; 1, a few to some stained immune cells; 2, many stained immune cells. MMP-7, -9, and -13 scoring were based on the intensity of cytoplasmic granular staining in hepatocytes and tumor cells, as follows: 0, no positivity in any cell; 1, staining in some cells; 2, staining in many cells. The scoring of stromal cells was based on the amount of stained immune cells, as follows: 0, no positivity in any cell; 1, some stained immune cells; 2, many stained immune cells.

**Figure 5 cancers-14-03396-f005:**
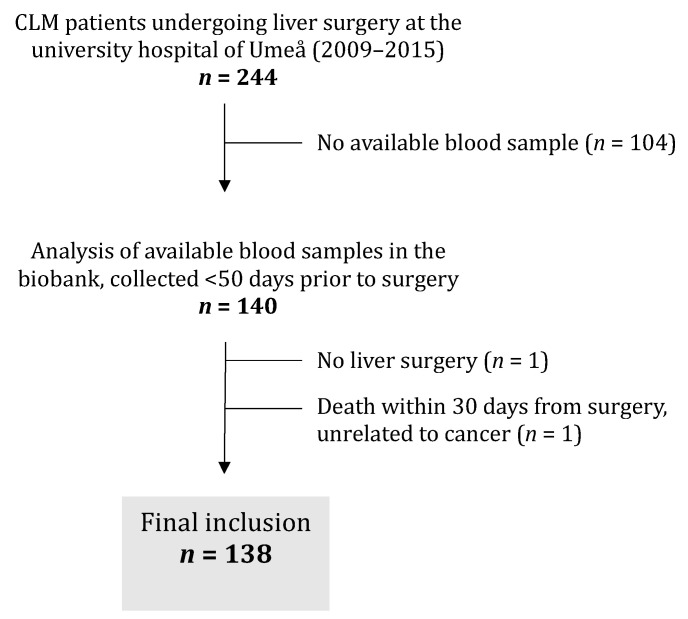
Flowchart of included and excluded patients in the plasma cohort. CLM = colorectal liver metastases.

**Figure 6 cancers-14-03396-f006:**
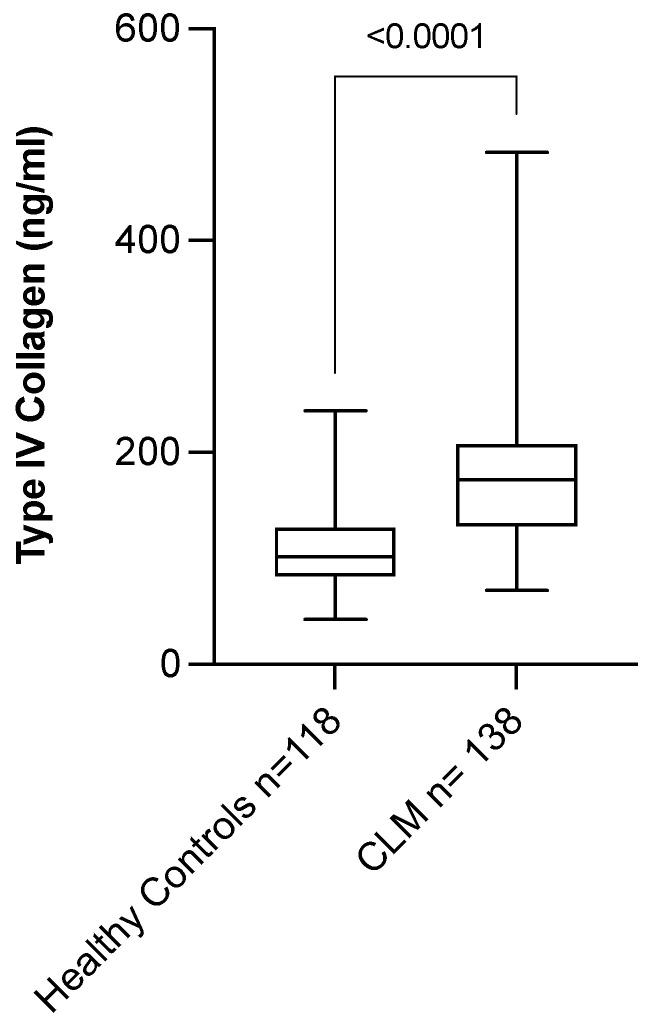
Levels of circulating type IV collagen (cCOL IV) in healthy controls and patients with colorectal liver metastases (CLM). Patients with CLM had significantly higher levels of cCOL IV (174.1 ng/mL (69.9–483.1)) compared with heathy controls (101.2 ng/mL (42.4–239.2)) (*p* < 0.0001).

**Figure 7 cancers-14-03396-f007:**
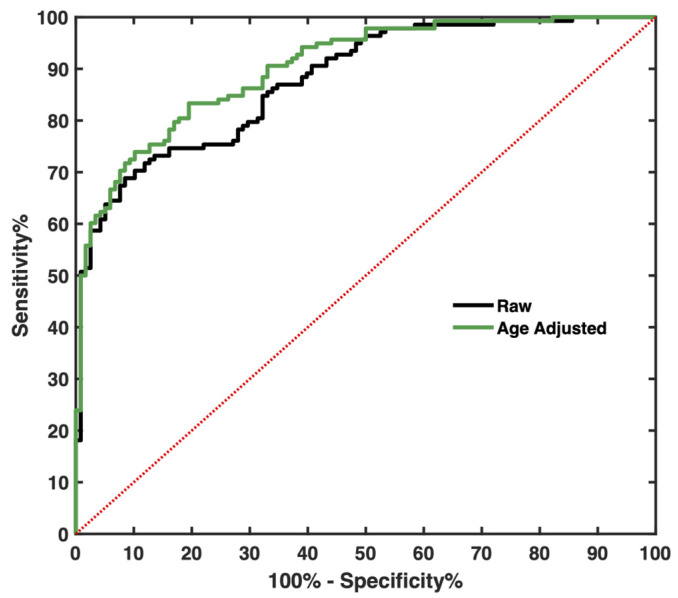
ROC curve analysis of circulating type IV collagen (cCOL IV) before (AUC 0.8791) and after adjusting for age (AUC 0.9027).

**Table 1 cancers-14-03396-t001:** Patient characteristics in the tissue cohort.

Characteristic	CLM Patients (*n* = 30)	CLM Patients for ISH Analysis * (*n* = 15)
Age CRC diagnosis (median, range)	62 (40–80)	68 (56–80)
Gender (M/F)	16/14	7/8
Primary tumor (colon/rectum/N.A.)	20/9/1	10/5
TNM stage		
I	3	3
II	6	5
III	8	3
IV	13	4
Chemotherapy		
Adjuvant	3	3
Preoperative chemotherapy	7	1
N.A.	1	0
Diagnosis metastasis		
Synchronous (<1 month)	13	4
Metachronous	17	11

CLM: colorectal liver metastases; ISH: in situ hybridization. * Tissue cohort (*n* = 15) for type IV collagen RNA-expression analysis by in situ hybridization. These patients are also included in the tissue cohort of *n* = 30.

**Table 2 cancers-14-03396-t002:** Median score of MMP * expression in colorectal liver metastases.

MMP Score 0–2	Immune Cells *n* = 30, *n* (%)	Tumor Cells *n* = 30, *n* (%)	Hepatocytes *n* = 30, *n* (%)
MMP-2			
0	1 (3.3)	25 (83.3)	1 (3.3)
1	20 (66.7)	5 (16.7)	5 (16.7)
2	9 (30)	0 (0)	24 (80)
MMP-7			
0	4 (13.3)	19 (63.3)	2 (6.7)
1	25 (83.3)	10 (33.3)	24 (80)
2	1 (3.3)	1 (3.3)	3 (10)
N.A.	0 (0)	0 (0)	1 (3.3)
MMP-9			
0	0 (0)	30 (100)	13 (43.3)
1	17 (56.7)	0 (0)	17 (56.7)
2	13 (43.3)	0 (0)	0 (0)
MMP-13			
0	8 (26.7)	30 (100)	3 (10)
1	17 (56.7)	0 (0)	22 (73.3)
2	5 (16.7)	0 (0)	5 (16.7)

* MMP: matrix metalloproteinase.

**Table 3 cancers-14-03396-t003:** Patient characteristics of the plasma cohort.

Characteristic	CLM Patients (*n* = 138)	Healthy Controls (*n* = 118)	*p*-Value
cCOL IV (ng/mL; median, range)	174.1 (69.9–483.1)	101.2 (42.4–239.2)	<0.0001
Age, blood sample (median, range)	66 (37–84)	60 (30–72)	<0.0001
Gender (M/F)	94/44	60/58	0.0049
Primary tumor (colon/rectum)	74/64		
TNM stage			
I	6		
II	15		
III	21		
IV	91		
N.A.	5		
N-positive primary tumor			
Yes	94		
No	37		
N.A.	7		
Primary tumor removed prior to liver surgery (no/yes)	39/99		
Liver surgery			
Liver resection	102		
Ablation *	13		
Resection + ablation *	23		
Preoperative chemotherapy			
Neoadjuvant	62		
Conversion therapy	20		
No treatment	53		
N.A.	3		
Liver metastasis > 1 (yes/no)	96/42		
Size metastasis > 5 cm (yes/no)	25/113		
Number of metastases (median, range)	2 (1–30)		
Extrahepatic metastases at resection			
Lung	4		
Carcinosis	6		
Lymph nodes	4		
Other	2		
Interval CRC–CLM ^+^, months (median, range)	0 (0–57)		

CLM: colorectal liver metastases. * Ablation refers to radiofrequency ablation or microwave ablation. ^+^ Time interval between primary CRC and detection of CLM.

**Table 4 cancers-14-03396-t004:** Specificity, sensitivity, and cutoff for cCOL IV.

Marker	I-Specificity	Sensitivity	Cutoff (ng/mL)
cCOL IV	16.1	74.6	135.2
cCOL IV (age-corrected)	19.5	83.3	unapplicable

## Data Availability

The data presented in this study can be shared in response to a reasonable request to the corresponding author.

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
