# Peer review of "Type IV Collagen in Human Colorectal Liver Metastases—Cellular Origin and a Circulating Biomarker"

_cancers, 2022, doi:10.3390/cancers14143396_

Round 1

Reviewer 1 Report

Well-known prognostic factors for CLM, such as number of metastases, lymph node metastases in the primary lesion, maximum diameter of metastases, tumor location, etc., are not prognostic factors in this study. This may be due to bias in the patients of this study or incorrect statistical methods. In Fig. 5, it is not possible to make a comparison because the clinical background is very different from that of healthy control.

Author Response

Comments and Suggestions for Authors
Well-known prognostic factors for CLM, such as number of metastases, lymph node metastases in the primary lesion, maximum diameter of metastases, tumor location, etc., are not prognostic factors in this study. This may be due to bias in the patients of this study or incorrect statistical methods. In Fig. 5, it is not possible to make a comparison because the clinical background is very different from that of healthy control.

Authors: Thank you for this remark. We did, however, not aim to investigate the known prognostic factors for outcome following liver surgery since this is well described. As stated, the aim was to analyze whether known prognostic factors affect the levels of circulating COL IV. The effect of prognostic factors on circulating levels of COL IV was investigated with Pearson’s correlation and t-test. We did observe a significant correlation between circulating levels of COL IV in CLM patients and the size of the largest metastasis, preoperative chemotherapy, and TNM stage.
Additionally, preoperative chemotherapy was related to the size of the largest metastasis and the TNM stage. Patients with high TNM-stage and large metastases are more likely to receive preoperative chemotherapy, explaining why these variables are related. Thus, we hypothesize that the correlation between circulating COL IV and these variables is due to a correlation between COL IV and tumor burden, which we have previously observed (Nyström H, Naredi P, Hafström L, Sund M. Type IV collagen as a tumor marker for colorectal liver metastases. Eur J Surg Oncol. 2011 Jul;37(7):611-7. doi: 10.1016/j.ejso.2011.04.010) albeit in a small patient cohort.

Regarding the comment concerning figure 5, it is correct that these were healthy controls. However, we believe this is an appropriate approach to investigating the potential of circulating COL IV as a tumor marker. The samples from these healthy controls were collected within the Västerbotten intervention program (VIP) (Norberg, M.; Wall, S.; Boman, K.; Weinehall, L. The Vasterbotten Intervention Programme: background, design and implications. Glob Health Action 2010, 3, doi:10.3402/gha.v3i0.4643), which is an intervention program aiming at reducing morbidity and mortality from cardiovascular disease and diabetes. As part of ordinary primary care, middle-aged persons (ages 40, 50, and 60) are invited to have their blood sampled to screen for risk factors. These blood samples are also available for other research purposes. We included 118 samples from the so-called VIP biobank in this study. We observed differences in age and gender between patients and controls, and age did affect COL IV levels in the controls. However, the case-control status had a more significant effect on COL IV levels. We also corrected for age.

Reviewer 2 Report

In the current manuscript entitled “Type IV collagen in human colorectal liver metastases– cellular 2 origin and a circulating biomarker”, the authors present an well-organized research in patients with colorectal liver metastases. However, there are some weaknesses that need to be addressed before this manuscript becomes acceptable for publication.

Firstly, although both the introduction and the discussion are well written, the authors should give more emphasis on the importance of studying cCOL IV in order to highlight the significance of their research. As for the applied methods, in situ hybridization was used for analysis of the RNA expression of COL IV. It would be interesting if the authors could perform qPCR for the estimation of the expression levels of COL IV in cancerous tissues in order to compare the results.

Figure 2 shows the immunofluorescence of CCD18 Co fibroblasts. However, the results for WST-1, the negative control for COL IV isn’t presented. Moreover, Figure 3 and table 2 present the results of  MMP expression in colorectal liver metastases. The authors could add an additional figure, for example a barplot, for an even better demonstration of their results.

Reviewer 3 Report

Type IV collagen in human colorectal liver metastases – cellular origin and a circulating biomarker

Lindgren and co-authors made a study with the samples from patients with colorectal liver metastases (CLM) collected from Umea university hospital. They aimed to investigate the cellular source of the COL IV and to evaluate the potential of circulating COL IV as a biomarker for the colorectal liver cancer. Authors have derived some critical conclusion that would benefit the field of cancer. However, I have few comments to make that would help to construct the manuscript better.

Simple summary needs English correction especially the second sentence.

Although authors had a wonderful opportunity to access a precious cohort of patient’s samples, major conclusions were drawn mostly from the histochemistry data. Why did authors not include any genomic or proteomic data in their study, which would have given deeper insight into the molecular mechanism underpinning the colorectal liver metastases. Moreover, 100+ plasma cohort with different groups of patients might have given opportunity to conduct molecular/biochemical work to elucidate the better picture of the disease mechanism. 

Figure 2 doesn’t have any significance and can be moved to supplements.

Figure 6 has the axis characters overlaid. So, I would recommend authors to correct it.

The last paragraph in the discussion section and conclusion can be merged.

Reviewer 4 Report

The publication from Lindgren M, et al aims to study the potential of cCOL-IV to be a potential biomarker of CLM.

Overall the study is clear and I only have minor comments in view to improve the paper.

-         - Simple summary: line 18 and line 20 should mention cCOL IV instead of only COL IV.

-         - The authors should write Cancer Associated Fibroblasts (CAF) rather than just fibroblast, when mentioning fibroblasts from CLM.  Tumor cells are able to rewire the fibroblast for their use that can explain the secretion of collagen IV.

-        -  Line 317: WS1 instead of WST1

The representation of the results can be improved:

- - Figure 1: can do a little box with bigger amplification of figure 1A. Difficult to see the RNA dots under this PDF.

-         - Figure 2: you should show all the staining of all tested cell lines. Furthermore, supplementary Fig 1 is quite weak, as the picture is out of focus and the nuclei is really small. Difficult to believe the result on 1 cell.

-         - Table 2: Supplementary table 1 should be put as main figure as It is easier to read. The current table is really difficult to interpret. Another way to present histological data is by bar graph as in figure 5B from DOI: 10.1073/pnas.1819303116 that can be more visual.

- Figure 4 and 6 seems to stack 2 same figure

Round 2

Reviewer 1 Report

There are still limitations in this research, though no room for improvement.

Reviewer 3 Report

Dear Authors,

Thanks for addressing the questions raised.